# Machine Learning and Swarm Optimization Algorithm in Temperature Compensation of Pressure Sensors

**DOI:** 10.3390/s22218309

**Published:** 2022-10-29

**Authors:** Hexing Wang, Jia Li

**Affiliations:** 1Institute of Microelectronics of the Chinese Academy of Sciences, Beijing 100029, China; 2University of Chinese Academy of Sciences, Beijing 100049, China

**Keywords:** pressure sensor, temperature compensation, differential evolution algorithm, quantum particle swarm optimization, dataset partitioning

## Abstract

The main temperature compensation method for MEMS piezoresistive pressure sensors is software compensation, which processes the sensor data using various algorithms to improve the output accuracy. However, there are few algorithms designed for sensors with specific ranges, most of which ignore the operating characteristics of the sensors themselves. In this paper, we propose three temperature compensation methods based on swarm optimization algorithms fused with machine learning for three different ranges of sensors and explore the partitioning ratio of the calibration dataset on Sensor A. The results show that different algorithms are suitable for pressure sensors of different ranges. An optimal compensation effect was achieved on Sensor A when the splitting ratio was 33.3%, where the zero-drift coefficient was 2.88 × 10^−7^/°C and the sensitivity temperature coefficient was 4.52 × 10^−6^/°C. The algorithms were compared with other algorithms in the literature to verify their superiority. The optimal segmentation ratio obtained from the experimental investigation is consistent with the sensor operating temperature interval and exhibits a strong innovation.

## 1. Introduction

Micro-electron mechanical systems (MEMSs) typically consist of mass-fabricable power sources, micromachines, micro-actuators and channels with feature sizes mostly on the order of micrometers [1,2,3]. Pressure sensors are one of the most widely used types of MEMS sensors that convert pressure signals received from the outside world into electrical signals and signal output [4,5], which have been widely used in consumer electronics, clinical medicine, autonomous driving and other fields. Compared with traditional pressure sensors composed of metal strain gauges, MEMS pressure sensors are produced based on semiconductor materials and processing technology, such as silicon, which allows these sensors to have the advantages of high sensitivity, low cost, miniaturization and easy integration. According to the working principle of pressure sensors, MEMS pressure sensors can be divided into piezoresistive, piezoelectric, capacitive, resonant and other types [6,7,8]. However, the generation of a piezoresistive effect needs to interact with external physical quantities, leading to an offset in output accuracy in daily work, which will be affected by the environmental temperature, atmospheric humidity, external pressure and other factors. How to improve the output accuracy of piezoresistive pressure sensors has become one of the topics of widespread concern in science and industry.

Currently, after extensive exploration, two common methods have been established to compensate for the sensor output: hardware compensation and software compensation [9]. Hardware compensation is a process of improving the accuracy of compensation for different drift scenarios through circuit design [10]. The circuit structure of the hardware compensation method is relatively simple, which has been widely used in engineering applications [11]. However, with the optimization and perfection of the packaging process, most of the sensors are put into use after the completion of the packaging, making it difficult to make subsequent adjustments after the completion of the hardware compensation circuit design. Furthermore, the change in the external temperature will have an irreversible influence on the components in the compensation circuit; thus, the error will tend to be significant with the growth of the service time [12]. Therefore, software compensation is gradually replacing hardware compensation in piezoresistive pressure sensors.

Software compensation is a process that involves improving the accuracy of piezoresistive pressure sensors by processing the collected output data as input by various algorithms [13]. According to the different types of algorithms, software compensation can be divided into parametric compensation and non-parametric compensation, based on intelligent optimization algorithms and machine learning [14]. Non-parametric compensation can solve the drawback that parametric algorithms cannot handle large-volume data, thus becoming the mainstream temperature compensation path [15,16,17]. The temperature compensation method based on numerical calculation has some limitations when the measurement accuracy of the pressure sensor is high. If the order of the fitting expression is higher than that of the least-squares fitting method for temperature compensation, an ill-conditioned problem will arise when solving the normal equation, meaning that a stable solution cannot be obtained. Machine learning models have powerful nonlinear approximation ability, overcoming some limitations of numerical calculation and providing a new choice for the temperature compensation of high-precision pressure sensors [18,19,20]. Suykens [21] introduced the least-squares support vector machine (LSSVM), which converts the inequality constraints in the traditional SVM to linear equations in the framework of regularization theory. Yao [22] optimized the wavelet neural network model and compensated the sensor temperature based on it. The results showed that the sensor temperature drift error can be optimized to 0.2%. Wang [23] proposed a temperature compensation method based on an improved cuckoo search optimizing a BP neural network for a multi-channel pressure scanner. The maximum full-scale error of all 32 channels is 0.02% FS (full-scale error) and below, which realizes its high-accuracy multi-point pressure measurement in a wide temperature range.

However, few algorithms are designed for sensors with specific ranges according to the existing research, and most methods ignore the operating characteristics of the sensors themselves [24,25,26,27]. This study considered this situation as an innovation point and proposes three temperature compensation algorithms, performing data acquisition and compensation for sensors with different ranges to investigate the range of the proposed algorithms, along with the optimal partitioning ratio. The results show that the intelligent optimization algorithms combined with the machine learning compensation method can improve the voltage output characteristics of the sensor and achieve higher compensation accuracy. Moreover, different compensation algorithms have different demands on the range of the sensor, which greatly improves the universality of the proposed algorithms. An innovative idea for algorithm compensation is provided through dataset segmentation based on the actual working environment temperature of the sensor.

## 2. Methods

### 2.1. Performance Parameters

Zero drift refers to the phenomenon in which the voltage across the bridge is not zero, primarily owing to the temperature change when the external stress of the pressure sensor is zero (i.e., at standard atmospheric pressure). The zero-drift coefficient is expressed as follows:(1)$$\alpha_{0}=\frac{U_{0}{}_{\max}-U_{0}{}_{\min}}{U_{FS}(T_{\max}-T_{\min})}$$
where *T*_max_ and *T*_min_ denote the maximum and minimum values of the external temperature, respectively, when the sensor is working, and *U*_0max_ and *U*_0min_ denote the maximum and minimum values of the output voltages of the sensor under no-load operation. A smaller zero-drift coefficient signifies that the output of the device is less significantly affected by the temperature and thus has superior output characteristics.

Sensitivity temperature drift refers to the phenomenon in which the voltage across the bridge increases nonlinearly as the input pressure increases when the operating ambient temperature rises or falls. The coefficient is defined as follows:(2)$$\alpha_{s}=\frac{U_{FS}{}_{\max}-U_{FS}{}_{\min}}{U_{FS\max}(T_{\max}-T_{\min})}$$
where *U_FS_* represents the output-voltage values at different temperatures under full-load operation. A smaller sensitivity temperature coefficient signifies that the output of the device is less significantly affected by the temperature and has superior output characteristics.

In this study, the zero-drift coefficient and sensitivity temperature coefficient were used to reflect the output accuracy of the sensor.

### 2.2. Differential Evolution Algorithm

The differential evolution (DE) algorithm is optimized using a genetic algorithm, which simulates the biological evolution process. It is based on a real number encoding strategy and improves the global search ability by providing the variation probability, and its variation mode is simple [28,29]. In addition, it features a real-time adjustable search strategy and improves the running speed of the algorithm.

The core steps of the differential evolution algorithm are the same as those of a traditional genetic algorithm, including mutation, crossover and selection. Before performing the mutation operation, the population must be initialized as follows:(3)$$P(X)=\{x_{j,i,g}|j=0,1,\ldots D;i=0,1,\ldots ,N_{p};g=0,1,\ldots ,g_{\max}\}$$
where *X* denotes the population individual, *x* indicates the individual gene, *j* indicates the chromosome number, *i* denotes the individual number, *N_p_* indicates the population size, *g* corresponds to the number of generations in the population and *g*_max_ indicates the maximum number of iterations, which, in this study, is considered to be 50. The difference vector is obtained by calculating all the genes of two random individuals in the population and acting on the third individual to complete the mutation operation, given the mutation probability. The least-squares error signal function was chosen in this study, which can be expressed as follows:(4)$$E_{k}=\frac{1}{2}{\sum_{j=1}^{o}\left\|y_{j}{}^{n\prime }-y_{j}{}^{n}\right\|}^{2}.$$
where *y* denotes the input sample vector, and *y′* indicates the output vector after passing through the multilayer perceptron.

The initial weights and thresholds of the BP algorithm are randomly generated, and when the size of the dataset increases, they may fall into local minima during the training. An optimal solution search can be performed using the DE algorithm prior to initializing the parameters to enhance the ability of the BP algorithm to perform a global search. The flow of the temperature compensation algorithm proposed in this study is shown in Figure 1, and the steps are detailed below.

(1) Firstly, the BP network weights and thresholds were initialized, and then they were encoded with inputs for the DE algorithm.

(2) Then, variation, crossover and selection operations were performed to achieve population evolution, and the least-squares error signal function was selected for the fitness function in this study.

(3) After the iteration was completed, the optimal weights and thresholds were obtained and used as the initialization parameters of the BP network. The learning rate was set to 0.01, and the momentum coefficient was set to 0.8. The BP network structure consisted of one input layer, three implicit layers and one output layer.

(4) The algorithm was executed 10 times to eliminate the randomness error of the BP network and then averaged to obtain the final output-voltage results.

### 2.3. Quantum Particle Swarm Optimization Algorithm

The quantum particle swarm optimization (QPSO) algorithm is optimized by PSO, where the particle displacement velocity is affected by both its historical and global optimal solutions [30,31]. In the PSO algorithm, the individual particle is affected by the local attractor, so the search result may fall into the local optimal solution and reduce the output accuracy of the algorithm. The QPSO algorithm retains the role of a local attractor and regards the search space as a quantum space. The position of particles appearing in space during each iteration follows the Schrodinger equation [32]. The Monte Carlo method can solve the probability of particles appearing in any position in space, and the solution results are as follows:(5)$$x_{i}^{g+1}=a_{i}^{g}\pm \frac{1}{2}L_{i}^{g}\ln(\frac{1}{r_{i}^{g+1}})$$
(6)$$L_{i}^{g}=\beta \left|C_{}^{g}-x_{i}^{g}\right|$$
(7)$$C^{g}=\frac{1}{N}\sum_{i=1}^{N}P_{ibest}^{g}$$
where *L* denotes the well length of the particle, *C* indicates the average of the optimal positions of all particles in space, *N* denotes the number of particles and *β* indicates the constriction factor. It can be seen that the coordinate points of the particles in the iteration are related to the average value of the optimal position of all particles in the QPSO algorithm, which weakens the potential energy effect of the local attractor and makes the algorithm more inclined to search for the overall optimal solution.

The algorithm can also improve the search efficiency for the overall optimal solution of the BP neural network. The flow of the temperature compensation algorithm is shown in Figure 2, and the steps are detailed below.

(1) The BP network weights and thresholds were initialized before they were encoded as inputs for the QPSO algorithm.

(2) The historical optimal value and global optimal value of particles were calculated, and then the optimal solution was found by comparing the fitness value after substituting in the fitness function.

(3) After the iteration was completed, the optimal weights and thresholds were obtained and used as the initialization parameters of the BP network. The structure of the BP network is the same as that of the DE algorithm.

(4) The algorithm was executed 10 times to eliminate the randomness error of the BP network and then averaged to obtain the final output-voltage results.

### 2.4. Fruit Fly Optimization Algorithm

The fruit fly optimization algorithm (FOA) is an intelligent optimization algorithm that integrates the biological characteristics of fruit flies into the algorithm. The core idea is the colony characteristics of fruit flies, which can plan the foraging path of the group using the concentration of the surrounding food [33,34]. Global optimization is achieved through interaction between different groups.

Based on the characteristics of the FOA, it can be combined with the least-squares support vector machine (LSSVM), which uses the output error signal to replace the relaxation variables and optimizes the inequality relations in the constraint conditions, converting them into equality relations in order to reduce the algorithm complexity and speed up the model solving speed. The flow of the temperature compensation algorithm is shown in Figure 3, and the steps are detailed below.

(1) The location of the fruit fly population was initialized using the Init function.

(2) The direction and distance of the fruit fly foraging were set, which can be expressed as follows:(8)$$\left\{\begin{array}{l} X(i,:)=X_{axis}+L_{x}\\ Y(i,:)=Y_{axis}+L_{y} \end{array}\right.$$
where *X*_axis_ and *Y*_axis_ denote the initial position of the fruit fly population, and *L_x_* and *L_y_* indicate the distances of horizontal and vertical movements.

(3) The distance formula *D* was used to calculate the distance of Drosophila from the initial position after flight, and the judgment value of the flavor concentration *S* was computed by taking the inverse, which are defined as follows:(9)$$\left\{\begin{array}{l} D(i,1)=\sqrt{X{\left(i,1\right)}^{2}+Y{\left(i,1\right)}^{2}}\\ D(i,2)=\sqrt{X{\left(i,2\right)}^{2}+Y{\left(i,2\right)}^{2}} \end{array}\right.$$
(10)$$\left\{\begin{array}{l} S(i,1)=\frac{1}{D(i,1)}\\ S(i,2)=\frac{1}{D(i,2)} \end{array}\right.$$

(4) The value of the taste concentration was substituted into the fitness function to obtain the individual taste intensity *S_i_* of the fruit fly.

(5) *S*_i_ and *S*_best_ were compared for different individual taste concentrations of fruit flies so that the initial position of fruit flies was updated. *X_axis_* and *Y_axis_* are defined as the penalty factor and kernel function parameters of the LSSVM, respectively.

(6) The optimal penalty factor and kernel function parameters were obtained through repeated iteration to proceed to the regression prediction.

(7) The algorithm was executed 10 times and then averaged to obtain the final output-voltage results.

### 2.5. Dataset Segmentation

When the scale of the calibration data is large, using all of them as the algorithm input will increase the complexity of the algorithm. Simultaneously, a small amount of data with large false fluctuations may reduce the compensation effect. Therefore, the dataset can be partitioned and divided into training and test sets based on different temperature intervals. If the temperature range is large during calibration, then some of the data within the operating temperature interval of the sensor can be used as the test set, whereas the remainder of the data can be used as the training set. This allows the piezoresistive sensor to determine the output compensation with high accuracy in the daily working environment temperature range. The splitting ratio is defined as follows:(11)$$\eta =\frac{\mathrm{Test}\;\mathrm{set}\;\mathrm{data}\;\mathrm{volume}}{\mathrm{Test}\;\mathrm{data}\;\mathrm{volume}}\times 100\%$$

Different segmentation ratios can cause the output accuracy of the sensors to differ. The optimal segmentation ratio is explored in this study.

## 3. Experiments

### 3.1. Instruments and Equipment

Three sensors with different ranges were selected for data calibration in this experiment: MSPD700-ASO (“Sensor A”), CD0302-350KP-A (“Sensor B”) and CD0302-70KP-A (“Sensor C”). The nonlinearity of each sensor was ±0.3% of the full-scale output. Since the packaged sensors were difficult to disassemble, the measurements were performed on bare pieces. The sensor models and their parameters are listed in Table 1.

The temperature test chamber is shown in Figure 4. The lower and upper limits of its operating temperature were −40 °C and 150 °C, which meet the requirements of data calibration. The temperature test chamber could be heated and cooled by setting the expected temperature for data calibration.

The experimental platform is shown in Figure 5. The external pressure transformer acted on the wafer through the air gun, with the pressure ranging from 0 to 1 MPa, which can reduce the relative error of the actual pressure within 0.01%. To read the output signal, the probe should be placed in contact with the varistor of the wafer in the probe stand. The output data reading device embedded in the experimental platform must be set to constant current source mode in the experiment.

The models and manufacturers of the instruments and equipment used in this experiment are shown in Table 2.

### 3.2. Data Calibration Scheme

Data calibration methods can be divided into static and dynamic calibration. Static calibration follows a given temperature change step, which is suitable for small-scale data calibration. The output data in dynamic calibration are measured in real time during the process of temperature increase or decrease, which is appropriate for large-scale data calibration. In this study, the static calibration method was utilized to investigate the pressure range of the sensor to which the three algorithms apply at temperatures ranging from −20 to 100 °C, with a calibration step of 20 °C. Subsequently, the dynamic calibration method was used to investigate the optimal partitioning ratio of the dataset, with a temperature range from −20 to 150 °C and a calibration step of 10 °C. The specific steps of the static calibration were as follows.

(1)The temperature test chamber was opened and left to stand for 2–3 min to allow its temperature to reach room temperature.(2)The initial temperature (−20 °C) and the initial input pressure (vacuum) were set. The output data were read after the temperature test chamber reached the set temperature. The given pressure change step according to the measuring range of the pressure sensors considered the algorithm complexity and the minimum change interval of the pressure gun. Data calibration was carried out after the pressure was changed and stabilized.(3)The operation in step (2) was repeated to carry out data calibration until the temperature rose to 100 °C and reduced to −20 °C.(4)To evaluate the effectiveness of the temperature compensation, each sensor was calibrated three times with the same conditions, and its repeatability was tested.(5)The output experimental calibration data were collated. For a specific sensor, the average output value of the same temperature in the process of heating and cooling was calculated as the output voltage at that temperature.

### 3.3. Data Calibration Results

Since the characteristics of the three sensors are basically the same, only the data calibration results of Sensor A are presented in this paper. The calibration results are shown in Table 3. There were eight pressure test points and seven temperature test points. U_p_ denotes the output voltage of the sensor. U_T_ indicates the output data of the temperature compensation resistor. Two-dimensional static calibration curves for Sensor A are shown in Figure 6. It can be observed from the figure that the degree of drift in the output curve increases as the input pressure increases.

The zero−drift coefficients and sensitivity temperature coefficients of the three sensors are listed in Table 4. Sensor B had the best calibration data performance, whereas Sensor C had the worst performance, which can be attributed to the fact that the sensor die has been compensated in the hardware. However, the different models have different compensation accuracies, which in turn leads to a difference in the calibration data. The drift coefficients of the output voltages of the different sensors were large, which demonstrates the necessity for software compensation.

### 3.4. Construction of Temperature Compensation Model

The temperature compensation model should be constructed first after the data calibration of the pressure sensor. From the description of the algorithms selected in this paper, it can be seen that different intelligent optimization algorithms are used to optimize the core parameters of the BP neural network and LSSVM. Therefore, the input of the BP neural network and LSSVM should be given to run the algorithm once the parameter optimization is completed.

As shown in Table 3, there were 56 calibration data (U_p_) and 7 output data of the temperature compensation resistor (U_T_). In the established model, the compensation pressure U_p’_ is taken as the output result in order to make the calculation results universal. Therefore, U_p_ and U_T_ can be regarded as the input array of the algorithm for three different machine learning algorithms. The input array is normalized to accelerate the training speed of the BP neural network and LSSVM. When the machine learning algorithm with optimized parameters is finished running, the output compensation pressure results are also normalized. Therefore, the final compensation pressure data should be obtained after inverse normalization of the output results. This paper takes the output results of Sensor A compensated by the DE–BP algorithm shown in Table 5 as an example. The output results U_p’_ under different calibration pressures were substituted into Equations (1) and (2) to calculate the compensated zero-drift coefficient and sensitivity temperature coefficient.

## 4. Selection and Optimization of Algorithm Parameters

### 4.1. Optimization of Parameters in DE

Before conducting temperature compensation based on the DE–BP algorithm, the variance probability M is optimized. In the early stage of the population iteration, the variation probability should be as high as possible to maintain the diversity of the population and prevent the global search performance from degrading toward a premature population maturity. In the late stage of the population iteration, an excessively high variation probability may cause the optimal solution to change once again, thereby increasing the search difficulty. Therefore, the probability of variation in the population is defined as follows:(12)$$\left\{\begin{array}{l} M=M_{0}3^{\lambda }\\ \lambda =e^{-\frac{g}{g_{\max}-g+1}+1} \end{array}\right.$$
where M_0_ is the coefficient of variation.

It can be found that when the population is in the early stage of evolution, the probability of variation tends to 3M_0_, which increases the population diversity. When the individuals in the population evolve to a later stage, the probability of variation decreases to M_0_, which is conducive to the retention of the optimal solution. In this study, experimental investigations were performed for the value of M_0_. It takes the value interval of [0, 1], and the step size is 0.2. The performance results of the algorithm for different M_0_ values are shown in Figure 7. From the figure, it can be observed that when M_0_ was 0.6, the algorithm had the highest optimal solution search efficiency. Therefore, M_0_ was set to 0.6.

### 4.2. Selection of Parameters in QPSO

In the solution result of QPSO, the shrinkage factor is used. If it is a constant value, the global optimal solution may be missed in the later stage of the iteration, reducing the convergence speed of the algorithm. Therefore, it can be defined as a decreasing function to make the algorithm easier to converge in the later stage. In this paper, three definitions were adopted to define the decreasing function [35,36], which can be expressed as
(13)$$\left\{\begin{array}{l} \beta_{1}^{}=k_{1}+(k_{1}-k_{0})\frac{g_{\max}-g}{g_{\max}}\\ \beta_{2}^{}=(k_{1}-k_{0})\times {\left(\frac{g}{g_{\max}}\right)}^{2}-(k_{1}-k_{0})(\frac{2g}{g_{\max}})+k_{1}\\ \beta_{3}^{}=(k_{0}-k_{1})\times {\left(\frac{g}{g_{\max}}\right)}^{2}+k_{1} \end{array}\right.$$
where k_0_ and k_1_ are constants. In this paper, k_0_ and k_1_ were set to 0.5 and 1.

The curves of the three decreasing functions are shown in Figure 8. It can be seen that their algorithm performances are similar. In order to reduce the complexity of the algorithm, β_1_ was chosen in this paper.

### 4.3. Optimization of Parameters in FOA

In the sensor temperature compensation algorithm based on the Drosophila optimization algorithm and LSSVM, a random number function is usually used when Drosophila moves in random steps after initialization, which has some defects. The quality of the search differs in diverse periods of the population iteration. It is closer to the global optimal solution in the middle and later periods of the iteration. If the population does not obtain the global optimal solution at this time, the global optimal solution may be missed when the random search step is large. Provided that the random search step size is small at the beginning of the algorithm iteration, it will reduce the search efficiency of the optimal value and affect the output accuracy of the algorithm. Accordingly, this paper redefined the moving step as
(14)$$\left\{\begin{array}{l} L=\lambda L_{\max}+L_{\min}\\ \lambda =e^{-\frac{g}{g_{\max}-g+1}+1} \end{array}\right.$$
where L_max_ denotes the maximum value of the moving step, which is set to 5, L_min_ indicates the minimum value of the moving step, g corresponds to the current iteration number, g_max_ denotes the maximal number of iterations and p is the regulatory factor, which is usually set to 1.

This paper conducted an experimental study on the value of L_min_, which varied from 0 to 0.02 under a step of 0.005. The performance results of the algorithm for different L_min_ values are shown in Figure 9. It can be seen that the optimal solution search efficiency of the algorithm was the lowest when L_min_ was 0, and the optimal solution search efficiency of the algorithm was the highest when L_min_ was 0.02. Therefore, L_min_ was set to 0.02.

## 5. Results and Discussion

### 5.1. Temperature Compensation Results Based on Calibration Data

The calibration data of different sensors were substituted into the different algorithms to perform temperature compensation, and the comparison of the results is shown in Table 6.

It can be seen from the table that the output accuracy of the three sensors was significantly improved after the compensation of the different types of algorithms, which reflects the advantages of combining the simulation of biological population characteristics with machine learning. The QPSO–BP algorithm had the smallest zero-drift coefficient in Sensor A compensation. The FOA–LSSVM algorithm had the optimal zero-drift coefficient value in Sensor B compensation. The DE–BP algorithm had the best performance in terms of the sensitivity temperature coefficient of the three sensors after temperature compensation, as well as the best training effect on the zero-drift coefficient of Sensor C. These compensation results show that different intelligent optimization algorithms may have different temperature compensation results for different sensor calibration data, which is caused by the randomness of traditional machine learning. In this experiment, the DE–BP algorithm achieved a better compensation effect than the other two intelligent optimization algorithms, and Sensor A was suitable for more algorithms. Therefore, the DE–BP algorithm was used to segment the dataset for Sensor A, as shown in the following. The comparison of the compensation results of different sensors under different algorithms is visually shown in Figure 10.

Table 7 intuitively shows the comparison of the zero-drift coefficient and sensitivity temperature coefficient between the proposed algorithms for parameter optimization and the temperature compensation algorithms presented in other studies. It can be seen that the three types of intelligent optimization algorithms that are combined with the machine learning framework obtained a higher output accuracy, and that the sensitivity temperature coefficients of three types of algorithms put forward in the paper are an order of magnitude lower compared with those of the other algorithms.

### 5.2. Temperature Compensation Results Based on Data Set Segmentation

In general, the typical operating environment temperature of the sensor is 20–100 °C; thus, the starting point of the test set data temperature was set to 20 °C, and the end point was set to 100 °C. If the test set is defined as data within the range from 20–−70 °C, then the segmentation ratio is 33.3%, as calculated according to Equation (11). After the segmentation ratio was given, the DE–BP algorithm was used for temperature compensation. To eliminate the randomness error of the BP network, the algorithm was repeatedly executed 10 times to perform averaging. Table 8 lists the zero-drift coefficient and sensitivity temperature coefficient of the sensor with different splitting ratios. Evidently, when the split ratio was introduced, the sensor output accuracy was significantly improved, and the zero-drift coefficient was reduced by two orders of magnitude, while the sensitivity temperature coefficient was reduced by three orders of magnitude, compared with the results of the uncompensated Sensor A calibration data.

Figure 11 shows the trend of the sensor drift coefficient with the split ratio. Apparently, the zero-drift coefficient had a minimum value of 4.29 × 10^−5^/°C at a split ratio of 33.3%. Correspondingly, the sensitivity temperature coefficient had a minimum value of 4.52 × 10^−6^/°C at a split ratio of 27.8%. When the splitting ratio was [22.2%, 44.4%], the output sensitivity temperature coefficient changed slightly. Therefore, in this study, 33.3% was considered the optimal value of the splitting ratio of the sensor input data, i.e., the calibration data between 20 and 70 °C were used as the test set, whereas the remainder of the data were used as the training set, and the output accuracy of the sensor was optimal.

## 6. Conclusions and Future Work

### 6.1. Conclusions

In this paper, we proposed three temperature compensation methods based on swarm optimization algorithms fused with machine learning for three different ranges of sensors and explored the partitioning ratio of the calibration dataset on Sensor A. The results show that the output accuracy of the three sensors was significantly improved after the compensation of different types of algorithms, which reflects the advantages of combining the simulation of biological population characteristics with machine learning. The optimal zero-drift coefficients of the three sensors were 7.37 × 10^−7^, 7.29 × 10^−7^ and 5.93 × 10^−7^, while the optimal sensitivity temperature coefficients of the three sensors were 7.76 × 10^−5^, 7.13 × 10^−5^ and 5.95 × 10^−5^. An optimal compensation effect was achieved on Sensor A when the splitting ratio was 33.3%, where the zero-drift coefficient was 2.88 × 10^−7^/°C and the sensitivity temperature coefficient was 4.52 × 10^−6^/°C. A comparison with other algorithms in other studies revealed that the intelligent optimization algorithm framework designed in this study achieves a higher output accuracy with reductions in one and two orders of magnitude in the zero-drift and sensitivity temperature coefficients, respectively. The experimental results are highly instructive for the software compensation method for piezoresistive pressure sensors. In addition, the introduction of the optimal splitting ratio allows the sensor to determine the output compensation with high accuracy in its daily operating ambient-temperature interval, which is innovative.

### 6.2. Future Work

In this paper, the content of the experiment still needs to be optimized. For different sensors, even if every step in the process is the same, the output parameters of the products on the same pipeline will have a certain degree of minor error. Therefore, for sensors with the same pressure range, several different products of the same batch should be selected for calibration, and the output results should be processed with the mean or optimal value. Due to the limitation of cost, the data calibration of this part will be carried out later.

## Figures and Tables

**Figure 1 sensors-22-08309-f001:**
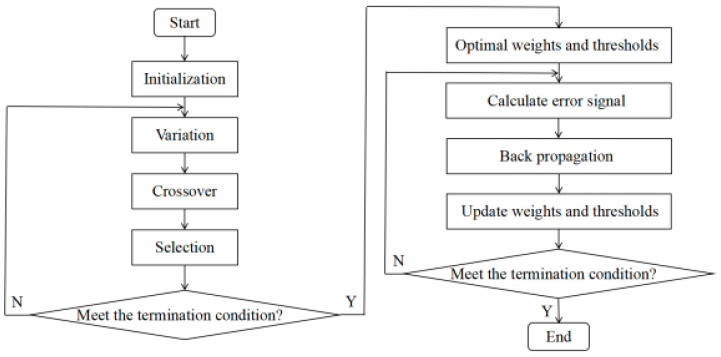
Temperature compensation flow using the DE algorithm.

**Figure 2 sensors-22-08309-f002:**
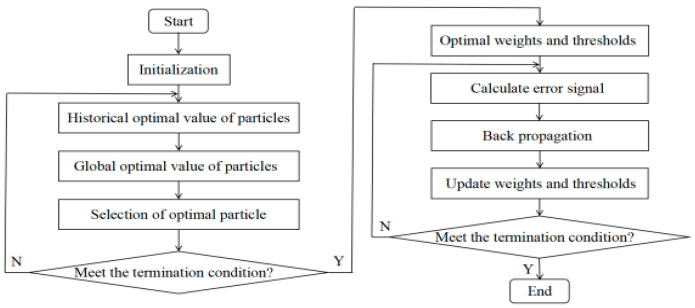
Temperature compensation flow using the QPSO algorithm.

**Figure 3 sensors-22-08309-f003:**
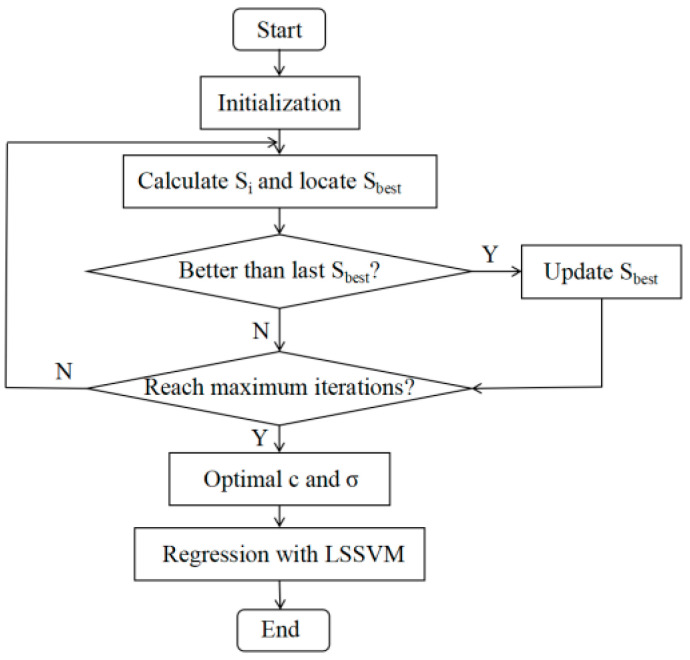
Temperature compensation flow using the FOA.

**Figure 4 sensors-22-08309-f004:**
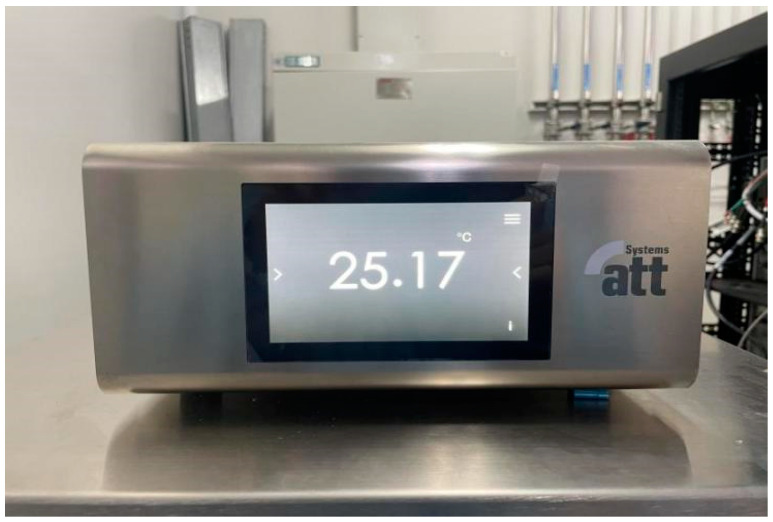
Temperature test chamber.

**Figure 5 sensors-22-08309-f005:**
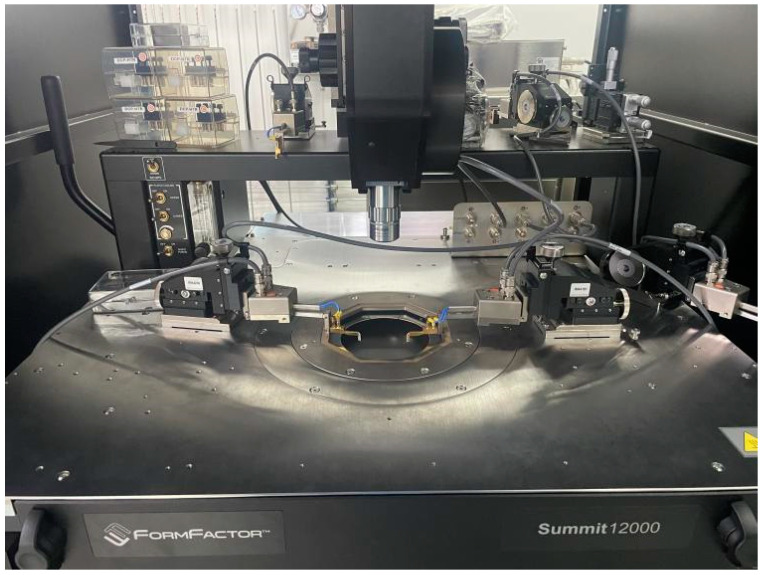
Experimental platform construction result.

**Figure 6 sensors-22-08309-f006:**
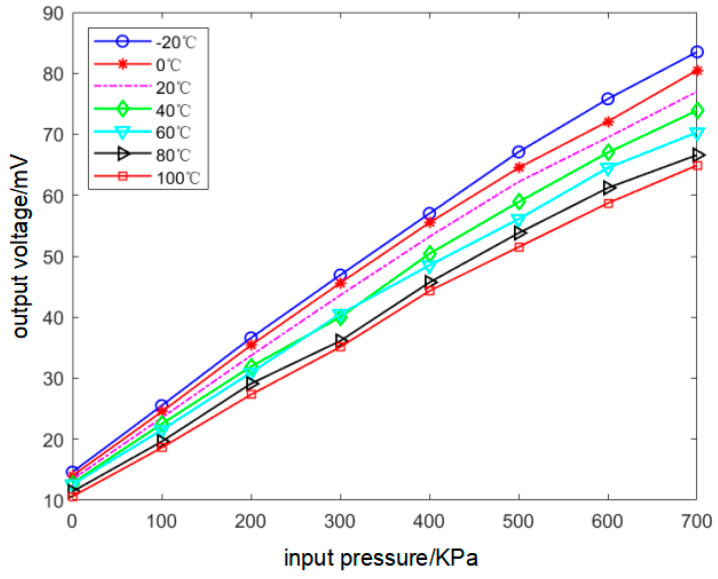
Output 2−D curves of Sensor A.

**Figure 7 sensors-22-08309-f007:**
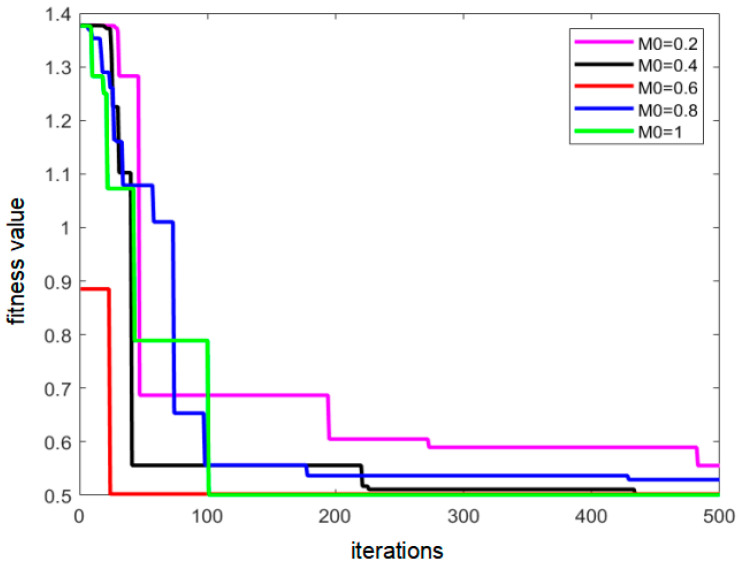
Schematic of algorithm performance under different *M_0_* values.

**Figure 8 sensors-22-08309-f008:**
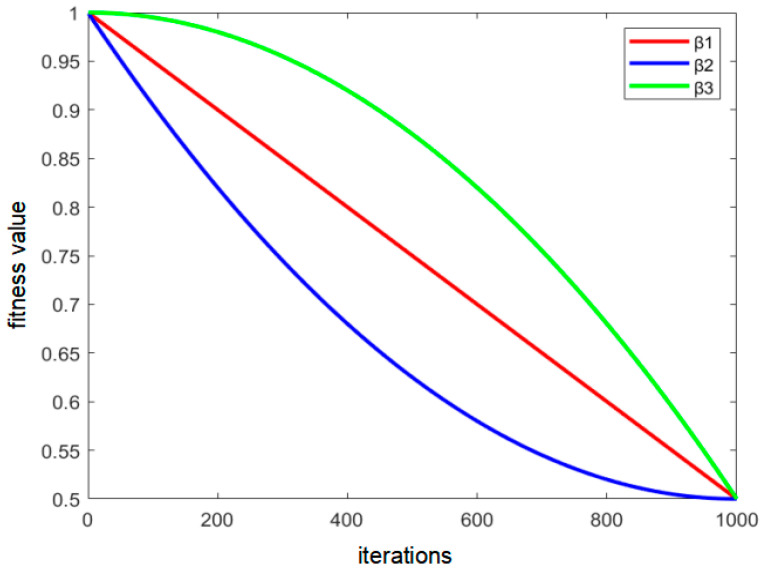
Diagram of the decreasing function.

**Figure 9 sensors-22-08309-f009:**
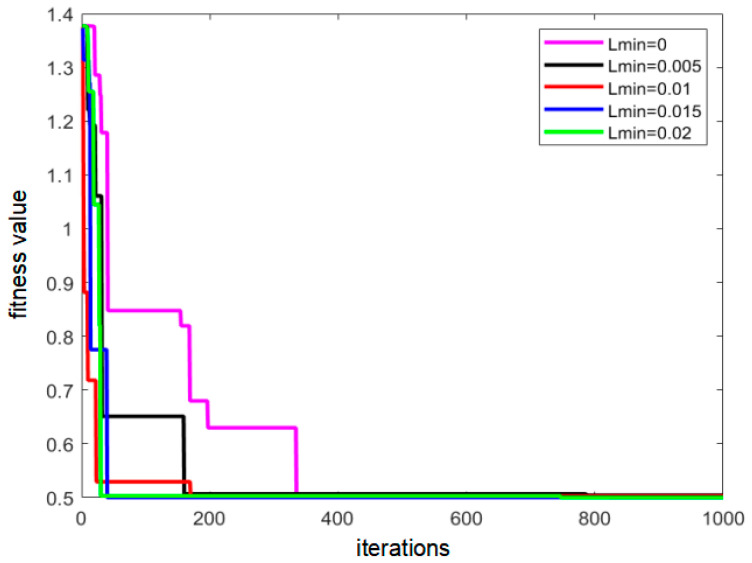
Schematic diagram of algorithm performance under different minimum *L_min_* values.

**Figure 10 sensors-22-08309-f010:**
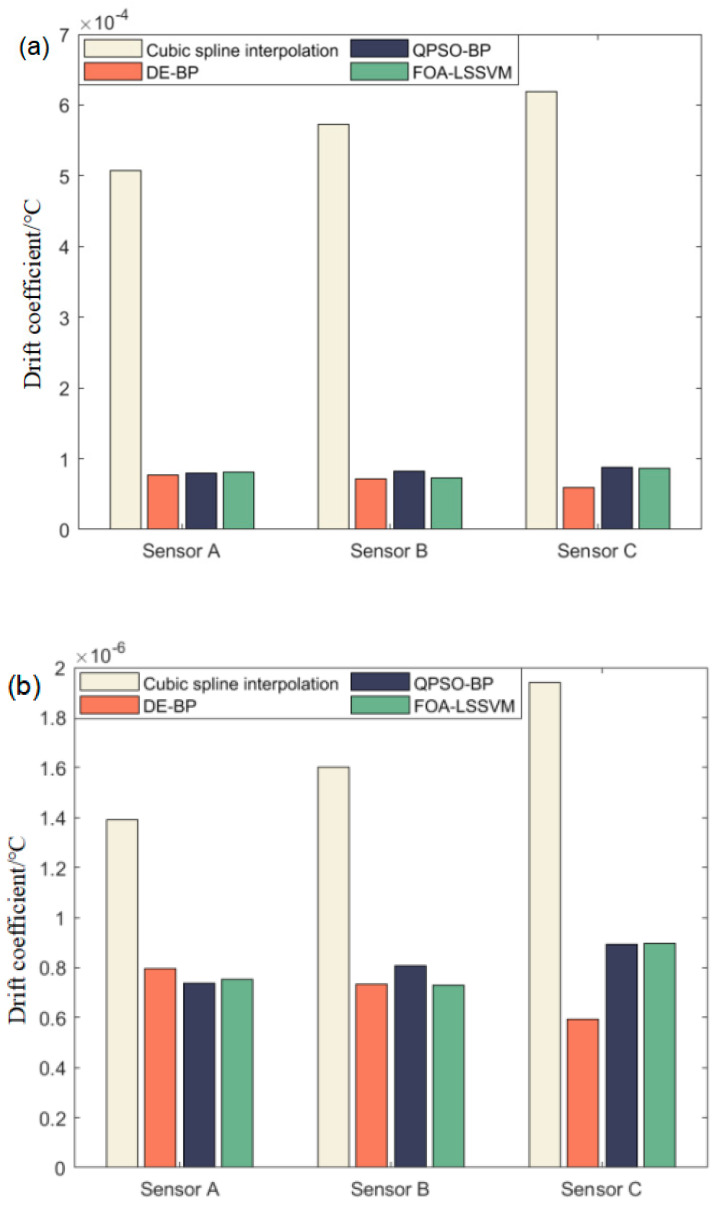
Comparison of different sensors: (**a**) zero-drift coefficient; (**b**) sensitivity temperature coefficient.

**Figure 11 sensors-22-08309-f011:**
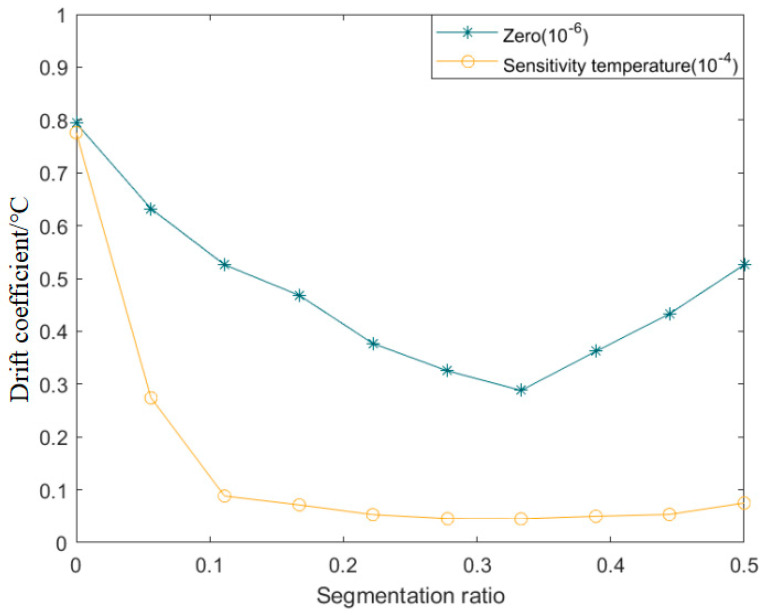
Trend of the sensor drift coefficient with different segmentation ratios.

**Table 1 sensors-22-08309-t001:** Sensor models and their parameters.

Sensor Model	Measurement Range (kPa)	Full-Scale Output (mV)	Operating Standard Temperature (°C)
Sensor A	700	100	20–100
Sensor B	350	90	
Sensor C	70	80	

**Table 2 sensors-22-08309-t002:** Models and manufacturers of the instruments and equipment.

Instruments and Equipment	Model	Manufacturer
Sensor A	MSPD700-ASO	Memsensing Microsystems
Sensor B	CD0302-350KP-A	FATRI Technologies
Sensor C	CD0302-70KP-A	
Temperature test chamber	Systems att C-Serials	Angelantoni test technologies
Air gun	Cascade-DPP210	FormFactor
Experimental platform	Summit-12000	

**Table 3 sensors-22-08309-t003:** Calibration results of Sensor A.

T/°C	U_T_/V	U_p_/mV							
		0 kPa	100 kPa	200 kPa	300 kPa	400 kPa	500 kPa	600 kPa	700 kPa
−20	2.734	15.822	27.715	39.868	51.081	62.051	73.053	81.189	90.960
0	2.956	15.246	26.749	38.503	49.610	59.616	70.340	78.521	87.714
20	3.220	14.637	25.680	36.744	47.558	58.022	67.892	75.844	84.052
40	3.481	14.064	24.724	34.926	45.849	55.005	64.325	73.158	80.622
60	3.823	13.680	23.257	33.487	43.827	52.343	60.589	69.685	75.907
80	4.149	12.590	21.603	32.055	41.864	50.377	59.250	67.476	73.394
100	4.395	11.483	20.172	29.598	39.075	47.833	55.626	63.391	70.044

**Table 4 sensors-22-08309-t004:** Drift parameters of sensors before compensation.

Sensor Model	α_0_ (°C)	α_s_ (°C)
Sensor A	4.03 × 10^−4^	1.94 × 10^−3^
Sensor B	3.99 × 10^−4^	1.86 × 10^−3^
Sensor C	4.04 × 10^−4^	2.41 × 10^−3^

**Table 5 sensors-22-08309-t005:** Temperature compensation results of Sensor A based on the DE–BP algorithm.

T/°C	U_p_/mV							
	0 kPa	100 kPa	200 kPa	300 kPa	400 kPa	500 kPa	600 kPa	700 kPa
−20	1.465	99.973	200.912	300.615	398.072	502.706	603.431	702.121
0	1.483	100.304	199.249	299.709	400.595	501.871	599.338	703.448
20	1.436	100.677	199.217	299.273	399.152	500.443	601.308	699.152
40	1.490	99.430	200.131	298.936	402.421	497.791	599.965	696.895
60	1.503	99.710	199.769	298.625	400.735	498.668	601.787	700.421
80	1.466	100.073	199.075	299.995	397.886	501.974	597.239	700.791
100	1.458	100.809	201.039	301.254	398.380	500.813	596.922	702.024

**Table 6 sensors-22-08309-t006:** Comparison of temperature compensation results based on different algorithms.

Sensor Model		Non-Compensation	DE–BP	QPSO–BP	FOA–LSSVM
Sensor A	α_0_/°C	4.03 × 10^−4^	7.94 × 10^−7^	7.37 × 10^−7^	7.51 × 10^−7^
	α_s_/°C	1.94 × 10^−3^	7.76 × 10^−5^	7.92 × 10^−5^	8.07 × 10^−5^
Sensor B	α_0_/°C	3.99 × 10^−4^	7.35 × 10^−7^	8.06 × 10^−7^	7.29 × 10^−7^
	α_s_/°C	1.86 × 10^−3^	7.13 × 10^−5^	8.21 × 10^−5^	7.29 × 10^−5^
Sensor C	α_0_/°C	4.04 × 10^−4^	5.93 × 10^−7^	8.93 × 10^−7^	8.98 × 10^−7^
	α_s_/°C	2.41 × 10^−3^	5.95 × 10^−5^	8.78 × 10^−5^	8.62 × 10^−5^

**Table 7 sensors-22-08309-t007:** Output accuracy of this paper’s algorithms compared with other algorithms in the literature.

Algorithm	α_0_ (/°C)	α_s_ (/°C)
Fuzzy neural network [37]	5.38 × 10^−6^	7.24 × 10^−4^
GSO–BP [38]	3.95 × 10^−6^	1.82 × 10^−4^
DE–SVM [39] DE–BP QPSO–BP FOA–LSSVM	4.03 × 10^−6^	1.03 × 10^−4^
	7.94 × 10^−7^	7.76 × 10^−5^
	7.37 × 10^−7^	7.92 × 10^−5^
	7.51 × 10^−7^	8.07 × 10^−5^

**Table 8 sensors-22-08309-t008:** Drift parameters of sensors at different segmentation ratios.

Segmentation Ratio	α_0_ (/°C)	α_s_ (/°C)
0	3.93 × 10^−4^	7.76 × 10^−5^
5.60%	1.05 × 10^−4^	2.74 × 10^−5^
11.10%	8.41 × 10^−5^	8.82 × 10^−6^
16.70%	7.23 × 10^−5^	7.13 × 10^−6^
22.20%	6.44 × 10^−5^	5.28 × 10^−6^
27.80%	5.30 × 10^−5^	4.52 × 10^−6^
33.30%	4.29 × 10^−5^	4.68 × 10^−6^
38.90%	4.86 × 10^−5^	4.97 × 10^−6^
44.40%	5.91 × 10^−5^	5.33 × 10^−6^
50.00%	7.17 × 10^−5^	7.49 × 10^−6^

## Data Availability

Not applicable.

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
