# Peer review of "Machine Learning and Swarm Optimization Algorithm in Temperature Compensation of Pressure Sensors"

_sensors, 2022, doi:10.3390/s22218309_

Round 1

Reviewer 1 Report

The text presents a description of three algorithm propositions for compensating for the effect of temperature on the response of piezo-resistive pressure sensors. Temperature changes affect the response of sensors of virtually any measured property. Therefore, the issue described seems to be very interesting. Unfortunately, the author's text is difficult to understand and contains various kinds of faults, both editorial and substantive. Therefore, I am obliged to suggest that the text be rejected in its present form and that the authors propose significant changes to it.

1) English language and style need improvement. The comment applies to virtually the entire text. In some places it is possible to get the meaning of the text, in others I simply did not know what the authors wanted to express.

2) The title of the text requires the addition of the information that the subject of the research is pressure sensors.

3) When the authors write [1,3], do they mean [1-3]? The way more than two positions are referred to needs improvement.

4) The description of the proposed algorithms is very generic. In general, adding descriptions of the operation of relatively well and widely known algorithms could be considered unnecessary.

5) The biggest problem, however, is that the descriptions relate poorly to the task they perform. (7) defines the so-called fitness function, but the terms used in the equation are not defined. In general, the description of all three methods lacks more detailed information on how these algorithms are actually implemented, what data is in the input and what is the effect of their operation.

6) If I have understood correctly (I could be wrong) each sensor must first be calibrated over a wide range of pressures and temperatures in order to use the proposed algorithms. What is the point of proposing such relatively complicated methods if, presumably, the same effect could be achieved by storing the calibration data in the sensor's memory and simply using it later?

7) The 'Instruments and equipment' section is missing some important details. Who is the manufacturer of the sensors? What equipment and with what measuring stand was used in the measurements? How temperature and pressure were changed, and how they were measured?

8) Showing the characteristics of all three sensors is probably unnecessary (figure 4). The characteristics of all sensors are similar. I think it is worth leaving the curves for one of the sensors.

9) Showing similar information in both, Figure 9 and Table 4, is a clear mistake. 

Author Response

Response Letter

Dear Reviewer,

We are very grateful to Reviewer for reviewing the paper so carefully. We have carefully considered the suggestion of Reviewer and make some changes.

Responds to the reviewers' comments:

  1. English language and style need improvement. The comment applies to virtually the entire text. In some places it is possible to get the meaning of the text, in others I simply did not know what the authors wanted to express.

We are very sorry for our problem of the English language and style. To polish up the language of the article, we select the Language Polish service on the MDPI website. The writing of the article is optimized on our own on the basis.

  1. The title of the text requires the addition of the information that the subject of the research is pressure sensors.

Thanks for your comment. The title in the original manuscript was indeed incomplete. So it is modified as follows: Machine Learning and Swarm Optimization Algorithm in Temperature Compensation of Pressure Sensors

  1. When the authors write [1,3], do they mean [1-3]? The way more than two positions are referred to needs improvement.

We are very sorry for our incorrect writing. [1,3] is updated to [1-3] at Line 28. This type of error in the article are rectified completely.

  1. The description of the proposed algorithms is very generic. In general, adding descriptions of the operation of relatively well and widely known algorithms could be considered unnecessary.

Thanks for your comment. Our description of the original algorithms did cause redundancy. Therefore, we simplify the description of the principle of the algorithms and put more emphasis on the establishment of the model in the process of modification. Some of the principles and formulas used by DE and QPSO algorithms are abbreviated.(from Line 116 to Line 189)

  1. The biggest problem, however, is that the descriptions relate poorly to the task they perform. (7) defines the so-called fitness function, but the terms used in the equation are not defined. In general, the description of all three methods lacks more detailed information on how these algorithms are actually implemented, what data is in the input and what is the effect of their operation.

Thanks for your comment. We are sorry for the lack of more detailed information on how these algorithms are actually applied on the temperature compensation model. So, we add a new section to explain how the algorithms used in the paper apply on the calibration date and what the input and output are from Line 343 to Line 377. The input of the BP neural network and LSSVM should be given to run the algorithm once the parameter optimization is completed. In the established model, the compensation pressure Up is taken as the output result in order to make the calculation results universal. Therefore, Up and UT can be regarded as the input array of the algorithm for three different machine learning algorithms. The input array is normalized to accelerate the training speed of the BP neural network and LSSVM. When the machine learning algorithm with optimized parameters is finished running, the output compensation pressure results are also normalized. The terms used in equation (4) are defined from Line 134 to Line 135. (original (7))

  1. If I have understood correctly (I could be wrong) each sensor must first be calibrated over a wide range of pressures and temperatures in order to use the proposed algorithms. What is the point of proposing such relatively complicated methods if, presumably, the same effect could be achieved by storing the calibration data in the sensor's memory and simply using it later?

Thanks for your comment. Each sensor must indeed be calibrated over a wide range of pressures and temperatures and some of sensors store the calibration data in the sensor's memory, which is called hardware compensation. However, with the optimization and perfection of the packaging process, most of the sensors are put into use after the completion of the packaging, making it difficult to make subsequent adjustments after the completion of the hardware compensation circuit design. Furthermore, the change in the external temperature will have an irreversible influence on the components in the compensation circuit; thus, the error will tend to be significant with the growth of the service time. The proposed methods of software compensation in the paper can improve these defects to some extent, compensating pressure sensors universally.

  1. The 'Instruments and equipment' section is missing some important details. Who is the manufacturer of the sensors? What equipment and with what measuring stand was used in the measurements? How temperature and pressure were changed, and how they were measured?

Thanks for your comment. We are sorry for the lack of some important details in the ‘Instruments and equipment’ section. The information of manufacture of the sensors and the equipment is added in Table 2 from Line 270 to Line 278. The experimental platform is shown in Figure 5. The external pressure transformer acted on the wafer through the air gun, with the pressure ranging from 0 to 1MPa. The specific steps of the static calibration are added from Line 290 to Line 304. The three algorithms apply at temperatures ranging from -20 to 100℃, with a calibration step of 20℃. The given pressure change step according to the measuring range of the pressure sensors considered the algorithm complexity and the minimum change interval of the pressure gun.

  1. Showing the characteristics of all three sensors is probably unnecessary (figure 4). The characteristics of all sensors are similar. I think it is worth leaving the curves for one of the sensors.

Thanks for your comment. We realize that showing the characteristics of all three sensors is indeed unnecessary. So we leave the curves for Sensor A and add its calibration data in Table 3.

  1. Showing similar information in both, Figure 9 and Table 4, is a clear mistake.

Thanks for your comment. We are sorry for this mistake and delete the Figure 9.

Reviewer 2 Report

Title: Machine Learning and Swarm Optimization Algorithm in Sensor Temperature Compensation

To my knowledge, the current paper does not provide much contribution in academic research. Instead, there might be some practical merit in industrial applications.

1. ML methods are introduced, but the authors fail to do justic to its necessity in the paper.

2. The decreasing function shown in Figure 6 is similar to the approach shown in https://www.tandfonline.com/doi/full/10.1080/03052150601047362

which needs to be discussed.

Author Response

Response Letter

Dear Reviewer,

We are very grateful to Reviewer for reviewing the paper so carefully. We have carefully considered the suggestion of Reviewer and make some changes.

Responds to the reviewers' comments:

  1. ML methods are introduced, but the authors fail to do justic to its necessity in the paper.

We appreciate it very much for this good comment. The necessity of machine learning methods in temperature compensation of pressure sensors has been added in the paper from Line 63 to Line 70. The temperature compensation method based on numerical calculation has some limitations when the measurement accuracy of the pressure sensor is high. If the order of the fitting expression is higher than that of the least-squares fitting method for temperature compensation, an ill-conditioned problem will arise when solving the normal equation, meaning that a stable solution cannot be obtained. Machine learning models have powerful nonlinear approximation ability, overcoming some limitations of numerical calculation and providing a new choice for the temperature compensation of high-precision pressure sensors.

  1. The decreasing function shown in Figure 6 is similar to the approach shownin https://www.tandfonline.com/doi/full/10.1080/03052150601047

362, which needs to be discussed.

We are very sorry for the issue. The main author of the paper consulted his senior fellow apprentice on the QPSO algorithm and applied the decreasing function. However, he didn’t affirm the source of the method. We asked the main author’s senior fellow apprentice and verified that the method utilized Li, Y.’s work, which has been added in the reference. We read the paper you gave carefully, finding that Decreasing-Weight Particle Swarm Optimization (DW-PSO) was addressed. Therefore, we contacted with Li, Y. and found that they utilized some conclusions in this paper. So we added the paper you gave in the reference too.

Reviewer 3 Report

Reviewer’s Comments

A major revision is being suggested for Manuscript No.: drones-1929885, titled, " Machine Learning and Swarm Optimization Algorithm in Sensor Temperature Compensation". The following are the observations:

1.      Introduction Section: The problem formulation is quite poor, author must add good quality literature to present the prior work in domain and to identify the research gap. Also, author needs to write the novelty of the work/contribution with reference to existing published work.

2.      Line 155: “The algorithm …….. optimal solution.”, rewrite the sentence.

3.      Line 158: “1) Initialized the …………. QPSO algorithm.”, rewrite the sentence and try to avoid multiple time and in single sentence.

4.      The writing / grammar and the tense is a major problem throughout the complete manuscript.

5.      Line 195: Author should check for the subscript and superscripts carefully and update.

6.      Line 289: (0,1] ?

7.      Author suggested to avoid unnecessary capitalization, suggested to check the same in the complete manuscript and update.

8.      In the conclusion the major findings with numbers should be provided.

I wish authors a great success.

Author Response

Response Letter

Dear Reviewer,

We are very grateful to Reviewer for reviewing the paper so carefully. We have carefully considered the suggestion of Reviewer and make some changes.

Responds to the reviewers' comments:

  1. Introduction Section: The problem formulation is quite poor, author must add good quality literature to present the prior work in domain and to identify the research gap. Also, author needs to write the novelty of the work/contribution with reference to existing published work.

Thank you for your comments. The introduction section of the original manuscript of our paper did not write the the prior work in domain and novelty clearly; In view of this, we have strengthened the introduction. We firstly write the necessity of machine learning. Three methods the prior came out in domain are presented from Line 70 to Line 79. From the research status, we find that existing methods don’t match the pressure sensors’ ranges and their working condition(especially environment temperature). Therefore, our work takes this as a starting point and conduct experiments on pressure sensors of different ranges. The novelty of the work has been rewrote from Line 80 to Line 92.

  1. Line 155: “The algorithm …….. optimal solution.”, rewrite the sentence.

Thanks for your comment. Our original writing was unclear, now we modify it as follows: The algorithm can also improve the search efficiency for the overall optimal solution of the BP neural network.(from Line 174 to Line 175)

  1. Line 158: “1) Initialized the …………. QPSO algorithm.”, rewrite the sentence and try to avoid multiple time and in single sentence.

Thanks for your comment. Our original writing was chaotic, now we modify it as follows: The BP network weights and thresholds were initialized before they were encoded as inputs for the QPSO algorithm.(from Line 177 to Line 178)

  1. The writing / grammar and the tense is a major problem throughout the complete manuscript.

We are very sorry for our problem of the writing and the tesne. To polish up the language of the article, we select the Language Polish service on the MDPI website. The writing of the article is optimized on our own on the basis.

  1. Line 195: Author should check for the subscript and superscripts carefully and update.

We are very sorry for our incorrect writing. Si is updated to Si at Line 214.

  1. Line 289: (0,1]

We are very sorry for our incorrect writing. (0,1] is updated to [0,1] at Line 397.

  1. Author suggested to avoid unnecessary capitalization, suggested to check the same in the complete manuscript and update.

Thanks for your comment. We have checked this problem in the manuscript and modify them.

  1. In the conclusion the major findings with numbers should be provided.

Thanks for your comment. We have added the major findingf with numbers from Line 555 to Line 557 as follows: The optimal zero-drift coefficients of the three sensors were 7.37E-07, 7.29E-07 and 5.93E-07, while the optimal sensitivity temperature coefficients of the three sensors were 7.76E-05, 7.13E-05 and 5.95 E-05.

Round 2

Reviewer 2 Report

Note:

The authors did not address the comments satisfactorily.

Reviewer 3 Report

The author has incorporated the suggested changes in the revised manuscript.  Now the manuscript can be accepted.